# Lightweight Federated Incremental Learning via Decoupled Replay

Xiuying Wang [1]   Yichen Li [2 3]   Hang Su [2]   Gaozhuo Liu [4]   Shiwei Li [2]   Chuang Zhao [5]   Jiangming Shi [6]
Imran Razzak [3]

## Abstract

Federated Incremental Learning (FIL) aims to learn streaming tasks across distributed clients without catastrophic forgetting while preserving privacy. Most existing methods mitigate forgetting by replaying historical samples, but it can pose privacy risks and incur high resource overhead, limiting deployment on resource-constrained edge devices. To address this challenge, we propose a novel Lightweight Federated Incremental Learning framework called **Li-FIL** that leverages dense features synthesized by a server-side secure generator to enable efficient feature-based decoupled replay. More specifically, each client extracts high-confidence features from new tasks, enrich them via mixup, and privatize them before uploading to the server, which reduces both storage and communication overhead. A generator is deployed on the server to learn the distributions of clients and generate global features for replay. Moreover, to enable clients to better learn from these dense features, we decouple local training into classifier stabilization and encoder regularization. This design allows feature replay and alignment between new and previous features to be conducted separately and more effectively. Extensive experiments demonstrate that Li-FIL outperforms other state-of-the-art methods by up to 10.14% in terms of accuracy on both old and new tasks with superior resource efficiency.

[1] International School, Beijing University of Posts and Telecommunications, Beijing, China [2] School of Computer Science and Technology, Huazhong University of Science and Technology, Wuhan, China [3] Department of Computational Biology, Mohamed bin Zayed University of Artificial Intelligence, Abu Dhabi, United Arab Emirates [4] School of Information and Communication Engineering, Beijing University of Posts and Telecommunications, Beijing, China [5] Tianjin University, Tianjin, China [6] Institute of Artificial Intelligence, Xiamen Univeristy, Fujian, China. Correspondence to: Yichen Li <ycli0204@hust.edu.cn>, Imran Razzak <imran.razzak@mbzuai.ac.ae>.

*Proceedings of the 43$^{rd}$ International Conference on Machine Learning*, Seoul, South Korea. PMLR 306, 2026. Copyright 2026 by the author(s).

## 1. Introduction

Federated learning (FL) is a machine learning approach that enables multiple distributed devices to collaboratively train a shared model without ever exchanging their private data (McMahan et al., 2017; Li et al., 2020; 2021; Hu et al., 2024). Recently, FL has been applied to extensive fields due to its good privacy protection, such as healthcare (Nasajpour et al., 2025), finance (Abadi et al., 2024), and telecommunications (Oladejo et al., 2025). However, conventional FL frameworks typically assume a static learning environment with a fixed set of tasks. In many real-world scenarios, data distributions are non-stationary, and new tasks arrive sequentially. When trained on a stream of new tasks, the global model often suffers from catastrophic forgetting (McCloskey & Cohen, 1989), hindering the deployment of FL in dynamic and evolving environments.

To tackle this challenge, Federated Incremental Learning (FIL) has emerged as the key paradigm by allowing clients to continuously learn from new tasks while preserving data privacy (Dong et al., 2023; Criado et al., 2022). GLFC (Dong et al., 2022) constructs a memory buffer on each local client and limits the local update. Re-Fed (Li et al., 2024b; 2025a) proposes coordinating clients to cache important samples based on both global and local perspectives for replay. The authors in (Qi et al., 2023) propose to use a generative network to reconstruct past samples for replay. Based on this, Target (Zhang et al., 2023) synthesizes data that simulates the global distribution of previous tasks for clients' exemplar-free knowledge distillation. CAN (Xuankun et al., 2025) leverages clients as navigators to guide both the synthesis process and the adaptive replay.

While these approaches have achieved significant performance improvement, sample-based replay methods suffer from two main limitations: (1) Historical data samples are often not permitted to be cached for replay or directly regenerated via generative models, which poses serious privacy concerns. In classic FL applications such as medical research, patient data is typically stored on secure local servers and subject to strict access regulations. Models are usually allowed only a single pass over such data, after which further access is prohibited (Suez et al., 2022; Shoer et al., 2023). Generating data via generative models in

such contexts can easily violate legal regulations (Smirnova & Travieso-Morales, 2024). (2) Although some existing methods propose selectively storing a portion of previous samples, it is difficult to strike a balance between buffer size and model performance. Many prior works assume relatively large memory buffers and commonly store up to 30% of historical data samples (Li et al., 2024d; 2025a). For unknown streaming tasks under edge devices, this assumption leads to a non-trivial storage overhead. Reducing the number of cached samples typically can cause a noticeable drop in model performance (Li et al., 2025b).

These limitations compelled us to explore more efficient paradigms, naturally leading our investigation to the low-dimensional feature space as an alternative. However, a closer analysis of existing feature-based strategies, such as feature distillation (Romero et al., 2014; Liu et al., 2020), uncovered that they fail to solve the fundamental storage and privacy issues—as they still require storing original data locally to generate features—and are further limited by a rigid reliance on instance-level matching that fails to preserve the structure of the feature manifold. This poses the second critical challenge: how to acquire a diverse set of features for replay in a manner that is both effective and privacy-preserving. The fundamental difficulty lies in generating features that are sufficiently representative of past tasks to prevent catastrophic forgetting, all while strictly avoiding local data storage to uphold user privacy.

In this paper, we argue for a feature-based replay strategy to achieve a lightweight yet effective FIL method. We propose Li-FIL, a framework leveraging the dense features to construct a decoupled replay mechanism for overcoming catastrophic forgetting. Specifically, the server will maintain a conditional generative model to synthesize feature representations of previous tasks for replay. This generator is trained on secure and dense feature–label pairs uploaded incrementally by clients, allowing it to accumulate knowledge from past tasks while preserving privacy. When learning a new task, clients receive the synthetic features from the server and implement a decoupled update strategy. For the classifier, we use direct feature replay with synthetic features to reinforce its stability. For the encoder, we introduce a structure-aware regularization technique to preserve the geometry of the feature manifold by aligning the neighborhood structures of new and synthetic features.

Through extensive experiments on various baselines and settings, Li-FIL demonstrates state-of-the-art performance, significantly mitigating catastrophic forgetting while reducing resource overhead. Our main contributions are summarized as follows:

- We systematically address the dual challenges of efficiency and data privacy in FIL, arguing that sample-based methods incur prohibitive resource costs and pri-

vacy risks, rendering them impractical for real-world edge deployment.

- To address this, we propose Li-FIL, a lightweight feature-based replay framework that reduces resource overheads while preserving privacy. By training a server-side conditional generative model on privacy-preserving features uploaded by clients and employing a decoupled client update strategy, Li-FIL effectively retains knowledge without storing or accessing raw historical data.

- We conduct extensive experiments on various datasets, demonstrating that Li-FIL achieves state-of-the-art performance and effectively addresses the challenges of privacy risks and resource overheads in FIL.

## 2. Related Work

**Federated Learning.** FL is a distributed machine learning paradigm where multiple clients collaboratively train a shared model, orchestrated by canonical algorithms like FedAvg that periodically aggregate locally trained parameters without exchanging private data (McMahan et al., 2017). To enhance its practicality, significant research has focused on optimizing communication efficiency through methods like model compression (Asheralieva et al., 2025) and quantization (Konečný et al., 2016; Liu et al., 2025), and bolstering privacy guarantees against model-based attacks (Gu et al., 2024) with differential privacy (Fu et al., 2024) and secure aggregation (Li et al., 2024a). Despite these advancements, most traditional FL frameworks operate under a static assumption, where the tasks and data distributions are fixed, failing to address the dynamic data streams.

**Incremental Learning.** IL aims to learn from a continuous data stream while preventing catastrophic forgetting (McCloskey & Cohen, 1989). This field encompasses three key scenarios (Van de Ven & Tolias, 2019), including Task-IL(Li & Hoiem, 2017), Domain-IL(Mirza et al., 2022), and Class-IL (Hou et al., 2019). Mainstream solutions are broadly categorized into three families: replay-based (Rebuffi et al., 2017; Feng et al., 2026; Qi et al., 2026), regularization-based (Kirkpatrick et al., 2017; Lu et al., 2025), and parameter-isolation methods (Rusu et al., 2016). While effective, these techniques are predominantly designed for centralized settings, and their direct application to federated environments is non-trivial due to privacy constraints and communication costs.

**Federated Incremental Learning.** FIL emerges at the intersection of FL and IL to enable collaborative model training on distributed, non-stationary data streams. The primary goal is to continually adapt a global model to new information at the client level while preserving previously acquired

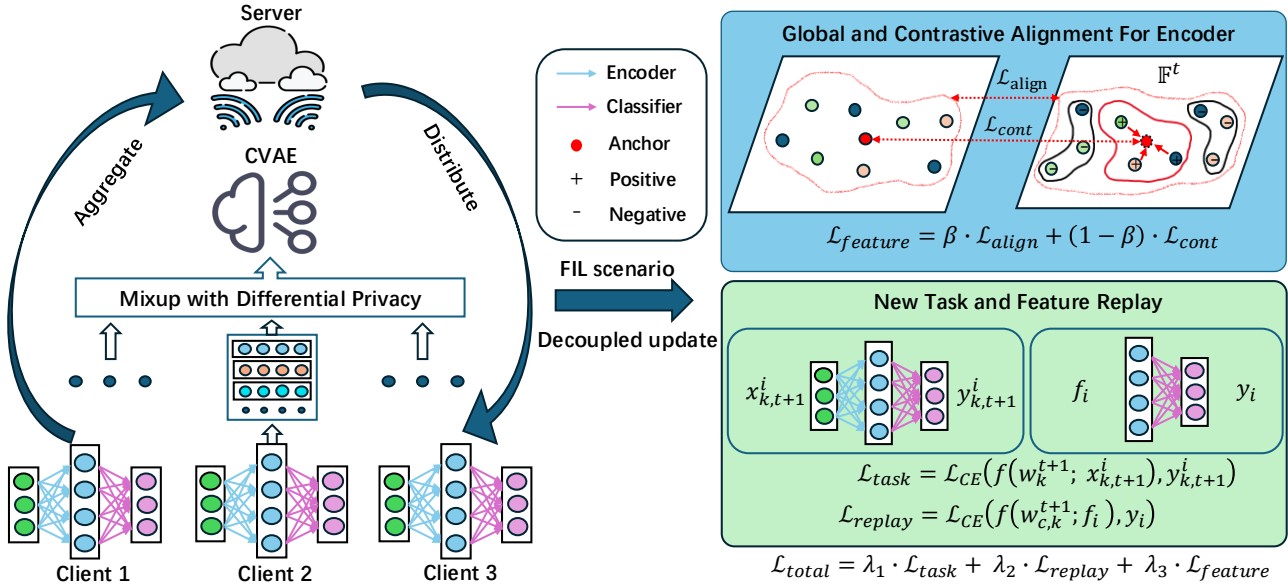

*Figure 1.* The Li-FIL Framework. Clients upload privacy-protected, mixed features to the server to train a central CVAE-based feature generator. When a new task arrives, clients download synthetic features generated by the server. They then perform a decoupled update strategy: (1) Classifier stabilization via direct replay loss on synthetic features, and (2) Encoder regularization via structure-aware feature distribution alignment loss between new task features and synthetic features, combining global and local alignment.

knowledge. Early explorations in this domain, such as the work by (Yoon et al., 2021), introduced methods for Task-IL scenarios that employ dedicated task IDs for different tasks to avoid interference. Another parameter isolation strategy, proposed by (Bakman et al., 2023), leverages orthogonal parameter subspaces for distinct tasks, effectively preventing new learning from overwriting past knowledge. Other approaches utilize generative models (Zhang et al., 2023; Qi et al., 2023; Mei et al., 2024; Xuankun et al., 2025) to synthesize data from past tasks for replay, thereby alleviating catastrophic forgetting without storing historical data (Li et al., 2024b; 2025a). More recently, to adapt regularization (Wang et al., 2024) for data heterogeneity, a method (Li et al., 2024c; Sun et al., 2025) was proposed that uses a personalized model to balance local and global knowledge when determining parameter importance. While effective, these methods can either impose prohibitive resource costs and privacy concerns through data-level replay or be too restrictive to effectively adapt to new knowledge. In contrast, our work focuses on developing a lightweight feature-based generative replay framework for FIL.

## 3. Methodology

This section details our proposed framework, Li-FIL, which is designed to address the challenging scenario of FIL. We begin by formally defining the problem setting, followed by a detailed breakdown of the framework's architecture and its core components.

### 3.1. Problem Formulation

The conventional FL setting involves $K$ clients collaboratively training a global model $w$ using their private datasets $D = \{D_1, D_2, \ldots, D_K\}$ without direct data sharing. Our work extends it to the FIL scenario, where local data evolves temporally. Each client $k$ sequentially encounters $t$ distinct learning tasks, $D_k = \{D_k^1, \ldots, D_k^t\}$. Each local task $D_k^t$ is associated with $N_k^t$ sample pairs $D_k^t = \{(x_{k,t}^{(i)}, y_{k,t}^{(i)})\}_{i=1}^{N_k^t}$, where $x_{k,t}^{(i)} \in X^t$ and corresponding label $y_{k,t}^{(i)} \in Y^t$. A key characteristic of this setting is that the label spaces for different tasks are disjoint, i.e., $Y^t \cap Y^j = \emptyset$ for $t \neq j$, which is a typical setup in Class-IL.

The core challenge in FIL is to update the global model $w^t$ to learn new classes from previously seen tasks $D^t = \sum_{j=1}^t \sum_{k=1}^K D_k^j$ across all clients, while preventing catastrophic forgetting of previous-task knowledge $\{D_k^j | j < t\}$. A critical condition constrains this process: when a new task arrives, clients lose access to all raw data from preceding tasks due to privacy regulations. We aim to find the optimal global model $w^t$ minimizing the empirical risk over all tasks $D^t$, formulated as:

$$\min_{w^t \in \mathbb{R}^d} \sum_{j=1}^t \sum_{k=1}^K \frac{1}{|D_k^j|} \sum_{(x,y) \in D_k^j} \mathcal{L}_{CE}(f(w_k^j; x), y), \quad (1)$$

$$\text{where } w^t = \sum_{k=1}^K \alpha_k^t w_k^t, \quad \alpha_k^t = \frac{N_k^t}{\Sigma_{j=1}^K N_j^t}.$$

where $\mathcal{L}_{CE}$ is the cross-entropy loss and $\alpha_k^t$ is the weight for

the global aggregation based on the number of data samples.

## 3.2. Li-FIL: A Feature-based Decoupled Replay Method

The core mechanism of our Li-FIL framework is a generative replay strategy that synthesizes secure and dense features for the decoupled local updating. In each communication round of Li-FIL, the server first generates synthetic feature representations of previous tasks using a conditional Variational Autoencoder (CVAE) (Sohn et al., 2015) trained on features from all clients. Then, each client will perform a decoupled model update. Specifically, each client trains its encoder and classifier separately using three loss terms: a task-specific loss on new data, a replay loss on the synthetic features to stabilize the classifier, and a structure-aware feature distribution alignment loss to regularize the encoder. After local training, each client extracts calibrated high-confidence features from the local new task, processes them via mixup with differential privacy noise to enrich the information density, and uploads the enhanced features to the server. These features are then used to update the global generator on the server.

The overall workflow is detailed in Appendix A, which presents the algorithmic procedure, while Fig. 1 illustrates the overall framework.

**Server-Side Feature Generation.** Considering the inaccessibility of previous samples and data privacy concerns, we propose to deploy a global feature generator on the server, which captures the local feature distributions from different clients.

For our decoupled replay to be effective, particularly for the classifier stabilization, this generator must be capable of producing a diverse set of features associated with specific class labels. We therefore employ a CVAE as our global feature generator. We specifically choose a CVAE for its training stability and label-guided generation capability, which allows the synthetic features to be constrained within a specified distribution space, while avoiding the training instability of GAN-based methods and the high computational overhead of diffusion models.

However, naively uploading all client features to train the CVAE would bring high communication costs and privacy risks and could yield a generator with poor capability, as outlier or low-quality features can degrade its training. To address this, we propose a two-stage feature contribution pipeline. First, a curation stage filters for high-quality features. For each new task $D_k^t$, client $k$ extracts a set of feature-label pairs $\{(f_i, y_i)\}_{i=1}^{N}$ from its local encoder $w_{e,k}^t$. To obtain a more reliable estimation of sample quality, we apply a temperature-scaled confidence score, which alleviates the issue that neural networks may assign overconfident predictions to misclassified samples (Guo et al., 2017).

Specifically, we compute

$$s_i = \max\left(\text{softmax}\left(\frac{f(w_{c,k}^t; f_i)}{T}\right)\right). \quad (2)$$

where $T > 1$ softly calibrates the logits and suppresses overconfident errors, and the local model is decomposed as a local encoder and classifier $w_k^t = \{w_{e,k}^t, w_{c,k}^t\}$. Samples whose calibrated confidence exceeds a threshold $\tau$ are regarded as high-quality data. We denote the selected high-quality feature set with size $N_c$ as:

$$\mathcal{F}_k^t = \{(f_j, y_j) \mid s_j > \tau\}_{j=1}^{N_c}. \quad (3)$$

Directly uploading these features to the server poses additional privacy risks: on one hand, they may be vulnerable to inversion attacks that attempt to recover the original data; on the other hand, the generator itself may inadvertently reveal sensitive information embedded in the learned representations. Therefore, we further adopt a mixup-based feature perturbation followed by a differential privacy (DP) mechanism to both reduce communication overhead and enhance privacy. We randomly divide $\mathcal{F}_k^t$ into $N_c/2$ disjoint pairs. For each feature pair in $\mathcal{F}_k^t$, we generate an interpolated feature-label pair:

$$\tilde{f}_i = \lambda f_a + (1 - \lambda)f_b, \quad \tilde{y}_i = \lambda y_a + (1 - \lambda)y_b, \quad (4)$$

where $\{(f_a, y_a), (f_b, y_b)\} \in \mathcal{F}_k^t$ and $\lambda \sim \text{Beta}(\cdot)$.

To bound sensitivity, each $\tilde{f}_i$ is clipped by its $\ell_1$-norm:

$$\tilde{f}_i \leftarrow \frac{\tilde{f}_i}{\max\left(1, \frac{\|\tilde{f}_i\|_1}{C}\right)}. \quad (5)$$

where $C = \kappa\sqrt{d}$ is the clipping threshold, and $d$ is the feature dimension. This ensures that the global $\ell_1$ sensitivity of the mixup function satisfies $\Delta_1 \leq 2C$. To guarantee record-level local differential privacy (LDP) for the replay channel, each clipped feature–label pair is privatized before transmission, which also improves robustness by preventing the generator from overfitting to any individual client's feature patterns. The privatized feature set is obtained by applying the Laplace mechanism to each clipped feature vector, yielding $\hat{\mathcal{F}}_k^t = \{(\hat{f}_i, \tilde{y}_i)\}_{i=1}^{N_c/2}$:

$$\hat{f}_i = \tilde{f}_i + \xi_i, \quad \xi_i \overset{\text{i.i.d.}}{\sim} \text{Laplace}\left(0, \frac{2C}{\epsilon}\right). \quad (6)$$

Under a standard record-level adjacency notion, this mechanism provides $\epsilon$-LDP for each uploaded feature, while model updates can be protected by standard FL mechanisms that are easy to integrate and orthogonal to our privacy-aware replay.

After each client $k$ uploads the secure and dense feature set $\hat{\mathcal{F}}_k^t$ to the server, the CVAE can be trained with all previous tasks $\hat{\mathcal{F}}^t = \Sigma_{j=1}^t \Sigma_{k=1}^K \hat{\mathcal{F}}_k^j$ with the following formulation:

$$
\mathcal{L}_{\text{CVAE}} = \sum_{(\hat{f}_i, \tilde{y}_i) \in \hat{\mathcal{F}}^t} \left[ \mathbb{E}_{q_\phi(\mathbf{z}|\hat{f}_i, \tilde{y}_i)} \log p_\theta(\hat{f}_i|\mathbf{z}, \tilde{y}_i) \right.
$$
$$
\left. - \text{KL}\left( q_\phi(\mathbf{z}|\hat{f}_i, \tilde{y}_i) \,\|\, p(\mathbf{z}|\tilde{y}_i) \right) \right]. \qquad (7)
$$

where the encoder $q_\phi(\cdot)$ maps each input feature–label pair to a latent distribution, while the decoder $p_\theta(\cdot)$ reconstructs the original feature conditioned on the latent variable and label. To guide the generator towards label-consistent and semantically meaningful representations, we use the class label $\tilde{y}_i$ as a conditional input to both encoder and decoder, enabling the synthesis of diverse yet class-controllable features. Operating solely on retained features, the CVAE incurs near-linear training cost, preventing the server from becoming a scalability bottleneck.

**Client-Side Decoupled Learning.** After collaboratively training a global generator to synthesize feature representations for previous tasks, we explore how to utilize them for better local learning to alleviate catastrophic forgetting. Our approach is motivated by the understanding that a model's components have distinct functional roles: the encoder learns to build a stable representation space for general features, while the classifier learns to form decision boundaries based on these features. As highlighted by studies on the anatomy of forgetting (Ramasesh et al., 2020), these components exhibit different vulnerabilities. Applying a uniform learning strategy to both is suboptimal, as it can render the encoder too rigid to adapt to novel concepts or fail to adequately protect the classifier's knowledge of previous tasks.

Based on this insight, we implement a decoupled replay strategy tailored to the distinct functional roles of the encoder and the classifier. Such decoupling allows each module to receive replay signals that are well aligned with its learning objective, leading to more stable and effective task transitions.

When the new $(t + 1)$-th task arrives, each client first receives a set of synthetic features with their corresponding labels, $\mathbb{F}^t = \{(f_i, y_i)\}_{i=1}^N$, where $N = \max(\{N_1^{t+1}, N_2^{t+1}, \ldots, N_k^{t+1}\})$. These features are generated by the server-side CVAE and carry consolidated knowledge of past tasks without exposing any raw data. To stabilize the classifier $w_{c,k}^t$, which is highly sensitive to decision-boundary drift and catastrophic forgetting, we perform direct feature replay using these synthetic samples.

The classifier is updated by minimizing

$$
\min_{w_{c,k}^{t+1}} \mathcal{L}_{\text{replay}} = \frac{1}{N} \sum_{i=1}^N \mathcal{L}_{\text{CE}}(f(w_{c,k}^{t+1}; f_i), y_i). \qquad (8)
$$

For the encoder $w_{e,k}^t$, traditional feature distillation is infeasible, as it would require access to past data, violating our core privacy and storage constraints. Moreover, instance-level matching is inadequate for incremental learning because the feature distributions of disjoint class sets often shift substantially across tasks. To overcome this issue, we propose aligning feature distribution between new and previous tasks by preserving both global and local structure in the feature space. First, to preserve the global structure, we align the overall distribution statistics using the squared Maximum Mean Discrepancy (MMD) loss:

$$
\min_{w_{e,k}^{t+1}} \mathcal{L}_{\text{align}} = \left\| \sum_{j=1}^{N_k^{t+1}} \frac{\phi\big(f(w_{e,k}^{t+1}, x_{k,t+1}^{(j)})\big)}{N_k^{t+1}} - \sum_{i=1}^N \frac{\phi(f_i)}{N} \right\|_{\mathcal{H}}^2 \qquad (9)
$$

where $\mathcal{L}_{\text{align}}$ computes MMD between the extracted features of the new task $D_k^{t+1}$ and the synthetic features of $\mathbb{F}^t$, and $\phi(\cdot)$ maps features into a reproducing kernel Hilbert space $\mathcal{H}$.

Beyond global distribution alignment, we further preserve the local topology of the feature space. This enables the encoder to maintain semantic consistency without relying on raw data or exact feature representations. It acts as a structure-aware regularization that mitigates representation drift and prevents negative transfer, thereby ensuring stable and generalizable representations in FIL settings. We introduce the contrastive alignment here. For each sample feature of the new task $D_k^{t+1}$, we select a positive set $\mathcal{P}_i$ consisting of its top-K nearest neighbors in the synthetic feature pool $\mathbb{F}^t$, and define the negative set $\mathcal{N}_i = \mathbb{F}^t \setminus \mathcal{P}_i$.

We then define a contrastive loss that encourages each sample feature of the new task to align with its positives while pushing it away from negatives. The contrastive alignment loss is given by:

$$
\min_{w_{e,k}^{t+1}} \mathcal{L}_{\text{cont}} = \frac{1}{N_k^{t+1}} \sum_{i=1}^{N_k^{t+1}} \log \Bigg[ 1 + \qquad (10)
$$
$$
\frac{\sum_{f^- \in \mathcal{N}_i} \exp\left( \text{sim}(f(w_{e,k}^{t+1}; x_{k,t+1}^{(i)}), f^-)/\tau \right)}{\sum_{f^+ \in \mathcal{P}_i} \exp\left( \text{sim}(f(w_{e,k}^{t+1}; x_{k,t+1}^{(i)}), f^+)/\tau \right)} \Bigg].
$$

where $\tau$ is a temperature parameter controlling the sharpness of the similarity scores. Finally, we combine both the global alignment and the local contrastive losses to synergistically update the local encoder $w_{e,k}^{t+1}$ as follows:

$$
\min_{w_{e,k}^{t+1}} \mathcal{L}_{\text{feature}} = \beta \mathcal{L}_{\text{align}} + (1 - \beta) \mathcal{L}_{\text{cont}}. \qquad (11)
$$

where $\beta$ is a hyperparameter that controls the trade-off between global distribution alignment and local structure preservation. Overall, the total training objective integrates task-specific learning, feature replay, and feature distribution alignment as three parts. The total objective is then:

$$\mathcal{L}_{\text{total}} = \lambda_1 \mathcal{L}_{\text{task}} + \lambda_2 \mathcal{L}_{\text{replay}} + \lambda_3 \mathcal{L}_{\text{feature}}. \qquad (12)$$

where $\mathcal{L}_{\text{task}}$ denotes the standard cross-entropy loss on the new-task data $D_k^{t+1}$, and $\lambda_1, \lambda_2, \lambda_3$ are non-negative hyperparameters that sum to 1 and control the relative weights of each loss term.

## 4. Experiments

### 4.1. Setup

**Datasets.** To evaluate our method, we use four datasets under federated incremental learning settings: **Fashion-MNIST** (Xiao et al., 2017), **CIFAR-10**, **CIFAR-100** (Krizhevsky et al., 2009), and **Tiny-ImageNet** (Le & Yang, 2015).

**Baselines.** For a fair comparison against other state-of-the-art methods, we configure the FIL tasks according to the protocols proposed by (McMahan et al., 2017; Rebuffi et al., 2017). We evaluate all methods using three categories of approaches: two representative FL methods: **FedAvg** (McMahan et al., 2017) and **FedProx** (Li et al., 2020); three memory buffer replay methods: **GLFC** (Dong et al., 2022), **ReFed** (Li et al., 2024b), and its improved variant **ReFed+** (Li et al., 2025a); three generative replay methods: **FedCIL** (Qi et al., 2023), **Target** (Zhang et al., 2023), and **CAN** (Xuankun et al., 2025).

**Configurations.** For a fair comparison, all methods use a ResNet18 (He et al., 2016) backbone. We simulate heterogeneous data distributions by partitioning the dataset among 20 clients using a Dirichlet distribution with parameter $\alpha$, where a lower $\alpha$ value signifies greater data heterogeneity. Unless stated otherwise, we use a default setting of $R = 150$ communication rounds and $E = 20$ local training epochs. In each round, we randomly select a fraction of clients ($k = 0.6$) to participate in the training. We experiment with different numbers of incremental tasks and each task arrives with new classes: 5/10/10 tasks with 2/10/20 new classes in each task for {Fashion-MNIST, CIFAR10}{CIFAR100} {Tiny-ImageNet}. For loss balancing, we set $\beta = 1/2$ in Eq. (11) and $\lambda_1 = \lambda_2 = \lambda_3 = 1/3$ in Eq. (12). These values were selected through grid search on a validation set and provided the best trade-off between stability and adaptability. We also found the method to be relatively insensitive to nearby values, with performance varying only within $\pm 0.5$–$1\%$. Our performance evaluation is based on Top-1 accuracy for all datasets except for Tiny-ImageNet, where we use Top-10 accuracy. We report the final accuracy

$A(f)$ upon the completion of the final streaming task, as well as the average accuracy $\bar{A}$ over all tasks. All results are reported as mean ± standard deviation over five random seeds.

### 4.2. Performance Overview

**Test Accuracy.** Table 1 summarizes the final accuracy $A(f)$ and average accuracy $\bar{A}$ across four heterogeneously partitioned datasets. The standard FL baselines obtain the lowest accuracies, with performance dropping sharply on CIFAR100 and Tiny-ImageNet. Memory-buffer methods offer clear improvements. Generative approaches further close the gap but still struggle as the number of classes increases. This limitation is often attributed to the declining quality of synthetic data, a consequence of challenges such as mode collapse in simple generators or the difficulty of training complex sample generators on limited data from a fraction of clients.

Li-FIL consistently outperforms all competing methods on every dataset up to $10.14\%$, demonstrating Li-FIL's robustness under the most challenging conditions.

**Resource Efficiency.** We evaluate the resource efficiency of Li-FIL by analyzing both convergence speed (Rounds) and computational cost (Time), with detailed results provided in Table 2. Li-FIL consistently requires the fewest communication rounds across all datasets, effectively reducing one of the primary bottlenecks in federated learning.

Li-FIL achieves lower per-round computation time than memory-buffer and generative baselines by using a lightweight server-side CVAE and feature-level replay instead of raw images. For example, CIFAR images contain 3,072 values ($32\times32\times3$), while ResNet18 features contain only 512 values, yielding a $6\times$ reduction in transmission cost, and up to $24\times$ for Tiny-ImageNet. This advantage increases with higher image resolutions, reducing communication overhead under bandwidth or latency constraints. The reduced input dimensionality also lowers model complexity: a CVAE trained on raw images has about 3.8M parameters, compared to 0.17M for a feature-level CVAE ($22\times$ smaller). By reducing per-round computational and communication cost, Li-FIL improves resource efficiency for practical FIL scenarios.

**Data Heterogeneity.** Figure 2 shows the effect of data heterogeneity controlled by the Dirichlet parameter $\alpha$. While all methods improve as data becomes more homogeneous, Li-FIL consistently maintains the largest margin, particularly under highly non-IID settings.

This resilience stems from Li-FIL's feature-based aggregation and replay mechanism. Clients contribute only calibrated high-confidence features, which, after mixup and DP

*Table 1.* Performance comparison of various methods with $\alpha = 1.0$ on four datasets.

| | Methods | Fashion-MNIST | | CIFAR10 | | CIFAR100 | | Tiny-ImageNet | |
|---|---|---|---|---|---|---|---|---|---|
| | | $A(f)$ | $\bar{A}$ | $A(f)$ | $\bar{A}$ | $A(f)$ | $\bar{A}$ | $A(f)$ | $\bar{A}$ |
| FL | FedAvg | 39.62±1.78 | 64.05±1.21 | 36.33±1.38 | 60.71±0.76 | 27.08±0.83 | 40.92±0.60 | 30.17±0.87 | 49.82±1.15 |
| | FedProx | 39.96±1.21 | 63.79±0.19 | 36.18±0.87 | 60.59±0.30 | 26.96±0.25 | 41.08±0.29 | 28.96±0.24 | 50.09±0.87 |
| Buffer | GLFC | 41.02±1.34 | 64.71±0.34 | 36.65±1.95 | 60.98±1.52 | 30.71±1.04 | 43.22±0.57 | 35.23±1.03 | 52.46±0.94 |
| | ReFed | 41.87±0.99 | 65.25±0.24 | 37.17±1.13 | 61.45±0.69 | 31.67±0.56 | 43.31±0.12 | 36.83±0.55 | 53.10±0.23 |
| | ReFed+ | 42.34±0.86 | 65.44±0.17 | 38.24±0.52 | 62.11±0.24 | 32.15±0.53 | 43.78±0.15 | 37.39±0.72 | 53.58±0.49 |
| Generative | FedCIL | 41.52±2.00 | 63.84±1.38 | 37.43±1.81 | 61.18±1.50 | 29.84±0.91 | 41.34±0.53 | 32.76±0.98 | 51.57±1.28 |
| | Target | 41.83±0.73 | 64.32±0.80 | 37.78±0.93 | 61.32±0.62 | 31.00±0.37 | 41.05±0.59 | 32.90±0.31 | 52.06±0.72 |
| | CAN | 42.20±1.14 | 66.76±1.26 | 38.46±1.54 | 62.13±1.34 | 31.79±0.79 | 42.03±0.72 | 34.12±0.65 | 52.36±1.06 |
| | **Li-FIL** | **44.39±1.53** | **67.91±1.09** | **40.29±1.03** | **62.56±1.11** | **35.61±0.82** | **45.73±0.67** | **40.31±0.49** | **55.31±0.61** |

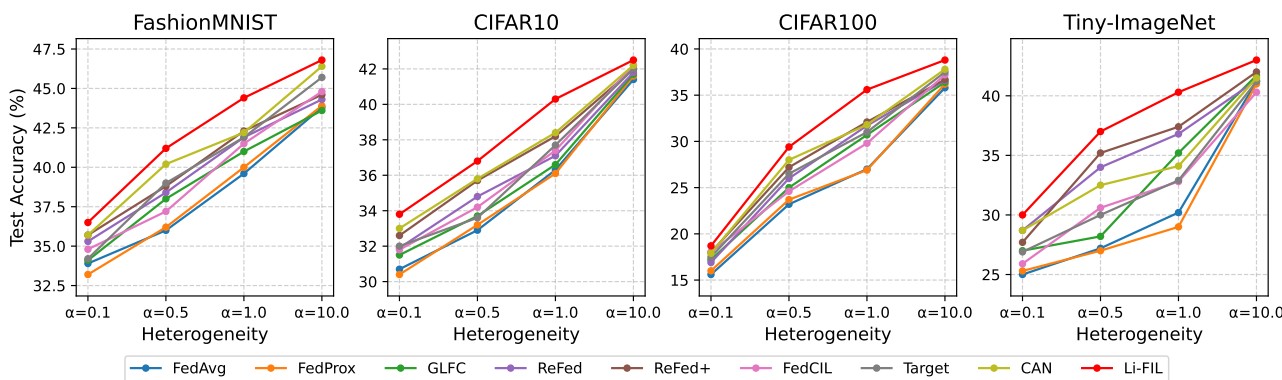

*Figure 2.* Performance of various methods with different levels of heterogeneity on four datasets.

noise, form a cleaner and more balanced pool for training the global CVAE. This allows the generator to capture a less client-biased feature distribution. Combined with decoupled replay, where classifier replay and encoder distribution alignment jointly mitigate decision-boundary drift and representation shift, Li-FIL produces reliable replay features even under extreme heterogeneity, leading to its consistently superior performance.

**Ablation Study.** As shown in Table 3, we conduct ablation studies to evaluate the contribution of each key component in Li-FIL. We consider five variants: (1) "-w/o mixup", which removes mixup from the client contribution stage; (2) "-w/o DP", which removes Laplace noise injection; (3) "-w/o $\mathcal{L}_{\text{replay}}$", which ablates the classifier stabilization loss; (4) "-w/o $\mathcal{L}_{\text{align}}$", which removes the global MMD-based alignment; and (5) "-w/o $\mathcal{L}_{\text{cont}}$", which removes the contrastive alignment component.

Overall, removing any component leads to performance

degradation, demonstrating the complementary nature of the proposed design. Eliminating mixup causes a noticeable drop, especially on CIFAR100, as the resulting client features become less dense and less informative, which degrades the quality of the learned CVAE and weakens feature replay. Removing DP noise yields only minor changes, indicating that Laplace perturbation preserves utility well. The injected noise not only provides privacy guarantees but also acts as a regularizer that prevents the generator from overfitting to specific client characteristics.

Ablating $\mathcal{L}_{\text{replay}}$ or $\mathcal{L}_{\text{align}}$ results in larger drops, confirming that classifier replay and global distribution matching are essential for mitigating decision-boundary drift and maintaining coherent representations across tasks. The largest decline appears when removing $\mathcal{L}_{\text{cont}}$, underscoring the importance of preserving local feature topology. Without this structure-aware constraint, the encoder experiences stronger representation drift, which is particularly detrimental on more fine-grained datasets.

*Table 2.* Efficiency comparison of various methods with $\alpha = 1.0$ on four datasets.

| | Methods | Fashion-MNIST | | CIFAR10 | | CIFAR100 | | Tiny-ImageNet | |
|---|---|---|---|---|---|---|---|---|---|
| | | Rounds | Time(s) | Rounds | Time(s) | Rounds | Time(s) | Rounds | Time(s) |
| FL | FedAvg | $251_{\pm 3.89}$ | 72.4 | $312_{\pm 3.14}$ | 90.8 | $620_{\pm 2.91}$ | 45.6 | $706_{\pm 1.54}$ | 53.2 |
| | FedProx | $230_{\pm 2.56}$ | 73.5 | $298_{\pm 2.05}$ | 92.9 | $631_{\pm 0.73}$ | 46.7 | $699_{\pm 1.08}$ | 56.4 |
| Buffer | GLFC | $245_{\pm 2.76}$ | 78.9 | $295_{\pm 1.59}$ | 101.2 | $597_{\pm 2.15}$ | 53.9 | $672_{\pm 2.67}$ | 65.4 |
| | ReFed | $237_{\pm 1.42}$ | 82.1 | $287_{\pm 1.03}$ | 110.7 | $585_{\pm 3.02}$ | 57.1 | $665_{\pm 2.31}$ | 70.3 |
| | ReFed+ | $234_{\pm 2.19}$ | 85.6 | $290_{\pm 2.92}$ | 121.5 | $583_{\pm 2.57}$ | 59.2 | $659_{\pm 1.75}$ | 71.9 |
| Generative | FedCIL | $231_{\pm 1.88}$ | 101.5 | $288_{\pm 2.54}$ | 134.4 | $589_{\pm 2.78}$ | 61.2 | $666_{\pm 2.08}$ | 73.5 |
| | Target | $239_{\pm 2.51}$ | 120.3 | $291_{\pm 3.12}$ | 142.9 | $593_{\pm 1.67}$ | 67.5 | $670_{\pm 1.62}$ | 74.2 |
| | CAN | $242_{\pm 1.04}$ | 115.8 | $283_{\pm 2.41}$ | 136.0 | $579_{\pm 1.55}$ | 66.7 | $664_{1.27}$ | 74.5 |
| | **Li-FIL** | $\mathbf{217_{\pm 2.93}}$ | 76.4 | $\mathbf{273_{\pm 2.07}}$ | 99.6 | $\mathbf{567_{\pm 2.34}}$ | 50.1 | $\mathbf{645_{\pm 2.45}}$ | 62.7 |

*Table 3.* Ablation Study of Li-FIL with $\alpha = 1.0$.

| Methods | CIFAR10 | | CIFAR100 | |
|---|---|---|---|---|
| | $A(f)$ | $\bar{A}$ | $A(f)$ | $\bar{A}$ |
| Li-FIL | $40.29_{\pm 1.03}$ | $62.56_{\pm 1.11}$ | $35.61_{\pm 0.82}$ | $45.73_{\pm 0.67}$ |
| -w/o mixup | $39.61_{\pm 1.10}$ | $61.47_{\pm 1.82}$ | $33.20_{\pm 0.84}$ | $44.04_{\pm 0.38}$ |
| -w/o DP | $40.25_{\pm 1.10}$ | $62.48_{\pm 1.17}$ | $35.49_{\pm 0.79}$ | $45.57_{\pm 0.70}$ |
| -w/o $\mathcal{L}_{\text{replay}}$ | $39.57_{\pm 1.23}$ | $61.75_{\pm 1.74}$ | $32.97_{\pm 1.06}$ | $43.78_{\pm 0.39}$ |
| -w/o $\mathcal{L}_{\text{align}}$ | $39.31_{\pm 0.71}$ | $61.65_{\pm 0.93}$ | $33.09_{\pm 0.90}$ | $43.33_{\pm 0.28}$ |
| -w/o $\mathcal{L}_{\text{cont}}$ | $38.12_{\pm 1.98}$ | $60.92_{\pm 1.41}$ | $31.52_{\pm 0.81}$ | $41.86_{\pm 0.47}$ |

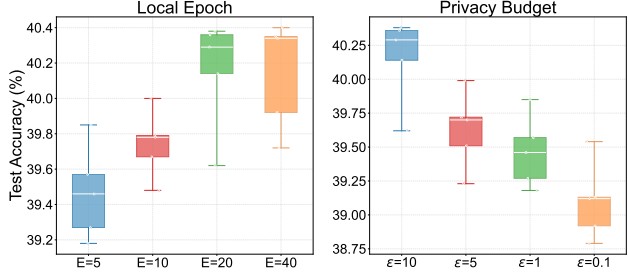

*Figure 3.* Parameter sensitivity analysis of local training epochs and privacy budget with $\alpha = 1.0$ on CIFAR10.

*Table 4.* Feature inversion attack results on CIFAR10.

| Methods | PSNR ↓ | SSIM ↓ | Attack Acc. (%) ↓ |
|---|---|---|---|
| Raw | $26.89_{\pm 3.27}$ | $0.59_{\pm 0.09}$ | $69.24_{\pm 1.52}$ |
| Mixup | $21.34_{\pm 1.63}$ | $0.43_{\pm 0.03}$ | $55.7_{\pm 1.78}$ |
| DP | $19.61_{\pm 2.16}$ | $0.38_{\pm 0.04}$ | $47.2_{\pm 2.39}$ |
| Ours | $\mathbf{10.93_{\pm 2.10}}$ | $\mathbf{0.29_{\pm 0.03}}$ | $\mathbf{29.3_{\pm 1.47}}$ |

**Privacy Evaluation.** We evaluate privacy leakage from the perspective of feature inversion. Specifically, we assess whether recognizable inputs can be reconstructed from the uploaded features by performing feature inversion attacks, with detailed configurations provided in Appendix B.

We report three standard metrics to evaluate feature inversion attacks: PSNR and SSIM to measure the visual similarity between reconstructed inputs and ground-truth images, and attack accuracy to quantify the success rate of recovering correct class labels from inverted samples. Higher PSNR and SSIM indicate stronger reconstruction quality,

while lower attack accuracy implies weaker semantic leakage. As shown in Table 4, our method consistently yields the lowest PSNR, SSIM, and attack accuracy, indicating significantly reduced visual and semantic leakage compared to raw features, mixup, or DP alone. This demonstrates that Li-FIL provides stronger privacy protection against feature inversion attacks. while qualitative reconstruction results are provided in Appendix C.1 for further illustration.

**Parameter Sensitivity Analysis.** Figure 3 presents a sensitivity analysis of Li-FIL with respect to key hyperparameters, with results aggregated over ten trials using standard boxplots. We observe that performance improves with an increased number of local training epochs, as clients have more opportunities to adapt to the new task. Conversely, the model's accuracy exhibits a clear degradation as the privacy budget $\epsilon$ is decreased. This result demonstrates the fundamental trade-off between privacy and utility inherent to DP mechanisms. A smaller $\epsilon$ enforces a stricter privacy guarantee, which, under the Laplace mechanism, necessitates an increase in the magnitude of the sanitizing noise. This additional noise, while crucial for privacy, can slightly

corrupt the feature representations, thereby impacting the model's final performance.

## 5. Conclusion

In this paper, we introduced Li-FIL, a novel and lightweight framework for Federated Incremental Learning that effectively addresses the dual challenges of data privacy and resource overhead inherent in existing methods. By using direct feature replay to stabilize the classifier and a structure-aware alignment loss to regularize the encoder on features synthesized by the central generator, Li-FIL effectively mitigates catastrophic forgetting while adapting to new tasks.

## Acknowledgments

This work is supported by the National Natural Science Foundation of China under grants 625B2073

## Impact Statement

This work presents Li-FIL, a privacy-aware and resource-efficient federated incremental learning framework that avoids storing or regenerating raw data by relying on protected features. It enables incremental learning in sensitive and resource-constrained settings, while requiring careful configuration to mitigate residual privacy risks.

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

# A. Algorithm

---

**Algorithm 1** Li-FIL: $t + 1$ task

---

**Require:** Model $w^t = \{w_e^t, w_c^t\}$, CVAE $\{q_\phi(\cdot), p_\theta(\cdot)\}$, $K$ clients with new task data $\{D_k^{t+1}\}$, $R$: communication rounds, $\eta$: learning rate, $E$: local epochs
**Ensure:** Updated model $w^{t+1}$ capable of classifying both new and old tasks

1: **for** round $r = 1$ to $R$ **do**
2:    **Server:**
3:       Sample client subset $\mathcal{S}_r \subset \{1, \ldots, K\}$
4:       Generate feature-label pairs $\mathbb{F}^t$ from past tasks using $p_\theta(\cdot)$
5:       Distribute $(w^t, \mathbb{F}^t)$ to clients in $\mathcal{S}_r$
6:    **Clients (parallel):**
7:    **for** each client $k \in \mathcal{S}_r$ **do**
8:       **for** epoch $e = 1$ to $E$ **do**
9:          Compute: task and decoupled loss (8)(11)
10:          Update model via total loss (12)
11:       **end for**
12:       Return updated model $w_k^{t+1}$
13:       **Feature Collection:**
14:       Extract high-confidence feature-label pairs from $D_k^{t+1}$, apply (4) (6) and upload.
15:    **end for**
16:    **Server:**
17:       Update generator $\{q_\phi(\cdot), p_\theta(\cdot)\}$ using (7)
18:       Aggregate: $w^{t+1} \leftarrow \sum_{k \in \mathcal{S}_r} \frac{|D_k^{t+1}|}{\sum_{j \in \mathcal{S}_r} |D_j^{t+1}|} \, w_k^{t+1}$.
19: **end for**

---

# B. Additional Configuration

## B.1. Setup

We evaluate the privacy-preserving effectiveness of our method through feature inversion attacks. The attack attempts to reconstruct original images from intermediate features extracted by the trained model. Specifically, we employ a gradient-based optimization approach where an initial random image is iteratively refined to minimize the distance between the extracted features and the target features. The reconstruction process uses Adam optimizer with a learning rate of 0.05 and runs for 3,000 iterations. The optimization objective combines feature reconstruction loss (MSE) with Total Variation regularization to encourage smooth reconstructions. Random spatial jittering is applied during optimization to improve robustness. The evaluation is performed using the ResNet18 model trained on the FIL task.

## B.2. Metrics

**Peak Signal-to-Noise Ratio (PSNR).** PSNR measures the ratio between the maximum possible signal power and the power of reconstruction noise. For image reconstruction, it is defined as:

$$\text{PSNR} = 20 \log_{10}\left(\frac{1}{\sqrt{\text{MSE}}}\right), \tag{13}$$

where MSE denotes the mean squared error between the reconstructed image and the ground truth. Higher PSNR values indicate better pixel-level reconstruction quality. However, PSNR primarily reflects pixel-wise differences and may not fully capture perceptual or semantic similarity.

**Structural Similarity Index (SSIM).** SSIM evaluates structural similarity by jointly comparing luminance, contrast, and structure between two images. It is computed as:

$$\text{SSIM}(x, y) = \frac{(2\mu_x \mu_y + C_1)(2\sigma_{xy} + C_2)}{(\mu_x^2 + \mu_y^2 + C_1)(\sigma_x^2 + \sigma_y^2 + C_2)}, \tag{14}$$

where $\mu_x$, $\mu_y$ denote local means, $\sigma_x$, $\sigma_y$ denote local standard deviations, $\sigma_{xy}$ is the cross-covariance, and $C_1$, $C_2$ are stabilization constants. SSIM ranges from 0 to 1, with higher values indicating stronger structural similarity. Compared to PSNR, SSIM provides a more perceptually meaningful assessment of reconstruction quality.

**Attack Accuracy.** Attack Accuracy measures the semantic leakage of reconstructed images by evaluating whether they can be correctly classified. It is defined as:

$$\text{Attack Accuracy} = \frac{1}{N} \sum_{i=1}^{N} \mathbb{1}\left[\arg\max f(w_{c,k}^t, \hat{x}_i) = y_i\right], \tag{15}$$

where $\hat{x}_i$ denotes the reconstructed image, $w_{c,k}^t$ is the classifier, $y_i$ is the ground-truth label, and $\mathbb{1}[\cdot]$ is the indicator function. Higher attack accuracy indicates stronger semantic preservation in reconstructed images and thus greater privacy leakage, while lower values suggest that the privacy-preserving mechanisms effectively prevent meaningful information recovery.

## C. Additional Results

### C.1. Feature Inversion Examples

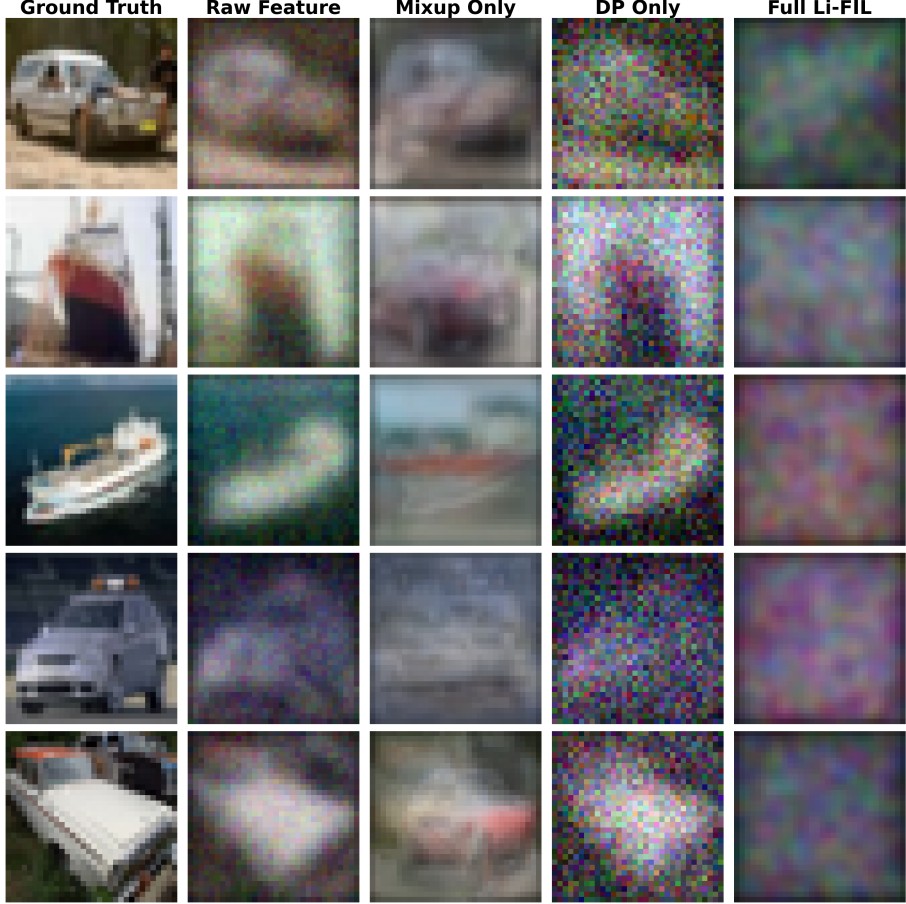

