# OpenReview forum: "Lightweight Federated Incremental Learning via Decoupled Replay"
_ICML.cc/2026/Conference — ICML 2026 regular_

### Official Review · Reviewer_MDXa · 2026-03-08

**Soundness:** 3
**Presentation:** 4
**Significance:** 3
**Originality:** 3
**Overall Recommendation:** 5
**Confidence:** 5

**Summary:**

The authors propose a novel Federated Incremental Learning (FIL) framework called Li-FIL, a feature-level generative decoupled replay method without storing or generating raw data to be privacy-preserving. Specifically, Li-FIL leverages a CVAE on the server trained on the curated clients' data to generate features for a well-designed decoupled replay mechanism. Empirical results show Li-FIL to be SOTA and privacy-preserving.

**Compliance With Llm Reviewing Policy:**

Affirmed.

**Final Justification:**

The rebuttal has addressed my concerns and the additional experiments have provided a much clearer understanding of the proposed method's advantages.

After reading the other reviewers' comments and the corresponding responses, I believe the authors have resolved most of the remaining issues. Consequently, I will increase my score.

**Key Questions For Authors:**

1. To better prove the effectiveness and scalability of the proposed method, can authors provide more results on parameter sensitivity and experiments with a larger number of participating clients as mentioned in Weaknesses 2 and 4?
2. To justify the choice of the CVAE, can the authors provide either theoretical insights or empirical comparisons with alternative generators
3. The current privacy evaluation mainly focuses on feature inversion attacks. Can the authors evaluate the method under other privacy attack settings to provide a more comprehensive assessment of privacy guarantees?

**Limitations:**

yes

**Strengths And Weaknesses:**

Strengths:
1. The paper is well structured in presentation and well-motivated to address the ongoing issues of privacy and dynamic scenarios in FIL.
2. The proposed method also provides an important and original insight of decoupling the model for imposing different levels of constraints to mitigate catastrophic forgetting.
3. The extensive experiments (e.g. ablation, parameter sensitivity) show Li-FIL's effectiveness across different settings. Additional results on feature inversion attack further justify that it is privacy-preserving.

Weaknesses:
1. Although the authors claim the method to be privacy-preserving and conduct an experiment of feature inversion, the privacy evaluation is still somewhat limited. More attacks (e.g. membership inference) should be conducted to better prove the privacy provided by the proposed Mixup with Differential Privacy (DP) mechanism.
2. Regarding the parameter sensitivity analysis, the paper only reports limited results. More detailed results should be demonstrated (e.g. beta in Eq. 11, lambdas in Eq. 12), which would help better understand how these parameters affect performance and improve the reproducibility of the method.
3. The choice of CVAE remains unclear. Recently, more advanced generative models like diffusion models have demonstrated strong performance, while simpler generators (e.g., lightweight MLP-based generators for reconstructing features) may also provide competitive performance with lower complexity. Maybe an explicit comparison with these generators either theoretically or empirically can justify the choice.
4. The number of participating clients in the experiments is relatively small. Evaluating the method with a larger number of clients would allow the scalability of the proposed approach to be better examined.

---

> ### Author Rebuttal · Authors · 2026-03-30
>
> **Response to Reviewer MDXa**
>
> We sincerely appreciate your review and insightful feedback. Below, we provide detailed, point-by-point responses to each of your comments.
>
> ---
> > **Q1&W2&W4.  More Results of Parameter Sensitivity and Scalability**
>
> **R1:** We thank the reviewer for raising this important point. We briefly describe the validation-based grid search performance in the **Configurations in Sec.4.1** and will include detailed sensitivity plots (e.g., varying λs and β) in the final version. From a design perspective, the equal weighting is motivated by the decoupled design of Li-FIL. Each term plays a complementary role: $L_{task}$ ensures plasticity on new tasks, $L_{replay}$ stabilizes the classifier, and $L_{feature}$ regularizes the encoder. Empirically, we found that balancing these objectives yields the best trade-off between adaptation and retention.
>
> Regarding scalability, we conducted additional experiments with 50 clients (30 sampled per round) on CIFAR10. The results show that Li-FIL maintains consistent performance gains over baselines, demonstrating its robustness to a larger number of participating clients.
>
> |Metric|FedAvg|GLFC|ReFed|Li-FIL|
> |:-:|:-:|:-:|:-:|:-:|
> |$A(f)$|29.13|31.75|31.49|32.34|
> |$\bar{A}$|50.27|53.47|54.91|55.53|
>
> ---
>
> > **Q2&W3. Justification of the Choice of CVAE**
>
> **R2:** Thank you for raising this point regarding the choice of CVAE. The choice of CVAE is therefore guided by a trade-off between **effectiveness, efficiency, and suitability** to the FIL setting. We systematically compare four common generators. (1) Diffusion models typically require **large-scale, clean, and static** datasets, whereas in our setting the server only has access to **limited, privatized, and incrementally collected** features. This makes it difficult for such models to be trained effectively and consistently across communication rounds. Moreover, their high computational cost and multi-step sampling process are not compatible with the lightweight and iterative nature of FL; (2) GAN-based methods can produce high-fidelity samples, but they are known to suffer from training instability and **mode collapse**, which is particularly problematic in our setting where capturing the diversity of feature distributions across tasks and clients is crucial for preventing forgetting; (3) simpler generators such as MLP-based models lack sufficient capacity to capture incremental complex feature distributions, leading to inferior replay quality.
>
> Overall, CVAE strikes a practical balance between modeling capability and system constraints, aligning with our goal of a lightweight and scalable FIL framework.
>
> ---
>
> > **Q3&W1.  Concerns about the Attack Mechanism**
>
> **R3:** We thank the reviewer for raising this important point. We agree that feature inversion alone may not fully characterize privacy leakage. To provide a more comprehensive evaluation, we further conduct a Membership Inference Attack (MIA) experiment on CIFAR-10, which captures a complementary aspect of privacy risk (as space is limited, detailed configuration will be updated in the supplementary).
>
> |Metric|Raw|Mixup|DP|Li-FIL|
> |:-:|:-:|:-:|:-:|:-:|
> |Attack Acc.(%)|71.1|68.8|60.3|57.2|
>
> Together with feature inversion, MIA provides a more holistic assessment covering both visual leakage and membership inference risk. As shown, while DP already serves as a strong theoretical defense, Li-FIL consistently achieves lower attack success rates, indicating that Li-FIL offers additional protection beyond DP alone.
>
> ---
> **Thank you again for the helpful feedback!**

---

> > ### Author Rebuttal · Reviewer_MDXa · 2026-04-02
> >
> > Thanks for the detailed response. The rebuttal has addressed my concerns and the additional experiments have provided a much clearer understanding of the proposed method's advantages.
> >
> > After reading the other reviewers' comments and the corresponding responses, I believe the authors have resolved most of the remaining issues. Consequently, I will increase my score.

---

### Official Review · Reviewer_kfBT · 2026-03-09

**Soundness:** 3
**Presentation:** 3
**Significance:** 3
**Originality:** 3
**Overall Recommendation:** 4
**Confidence:** 5

**Summary:**

This paper proposes Li-FIL, a lightweight FIL framework that mitigates catastrophic forgetting while preserving privacy and improving efficiency. The key idea is to perform feature-level replay using a server-side generator trained on privatized client features. Clients upload curated features that are processed with mixup and differential privacy before being used to train a conditional VAE. During incremental learning, synthetic features are directly replayed to stabilize the classifier, while the encoder is further regularized via L_align and L_cont.
The effectiveness of Li-FIL is verified empirically through experiments on several benchmark datasets.

**Compliance With Llm Reviewing Policy:**

Affirmed.

**Key Questions For Authors:**

Can authors

(1) Provide a comparion with generator-free methods,

(2) Include more attack mechanism, and provide a formal privacy guarantee, and

(3) Systematically analyze the communication overhead.

I'll revise my score if the authors can address my concern.

**Limitations:**

yes

**Strengths And Weaknesses:**

Strengths:

(1) The proposed method is well designed and demonstrates strong effectiveness, where the privacy guarantee is furter verified empirically.

(2) The authors effectively combined existing tricks like mixup with the proposed decoupled feature-level replay mechanism.

(3) Addressing the resource constraints in FIL tasks for edge deployment is an important research direction.

Weaknesses:

(1) CVAE is the core of the proposed method, but its scalability remains unclear, as in the setting, there are 200 classes at most. Some generator-free methods collect the features uploaded to calculate a prototype for each class and use some random synthesis data based on the prototypes for replay, which is more lightweight (without training any generators).

 (2) Feature Inversion alone may not fully characterize privacy leakage in FL. Moreover, although differential privacy noise is applied to the uploaded features, the paper does not provide a formal analysis of the resulting (ϵ,δ)-DP guarantees for the overall system.

(3) Communication overhead can be systematically analyzed, which is a critical factor in FIL, as the number of uploaded features is unstable.

---

> ### Author Rebuttal · Authors · 2026-03-30
>
> **Response to Reviewer kfBT**
>
> We sincerely appreciate your review and insightful feedback. Below, we provide detailed, point-by-point responses to each of your comments.
>
> ---
> > **Q1&W1.  Comparison between Generator-free Methods and Li-FIL**
>
> **R1:** Thank you for raising this point regarding the necessity and scalability of CVAE. We agree that prototype-based approaches offer a lightweight alternative by only collecting prototypes (e.g., class means) without generators. However, such methods inherently approximate each class with a single or limited set of points, which restricts their ability to capture the underlying feature distribution and intra-class variation, especially under non-IID incremental settings. In contrast, our CVAE-based generator models the **richer conditional feature distribution**, enabling the synthesis of diverse feature samples for each class.
>
> To support this point, we additionally compare Li-FIL with FCLPF[1], a SOTA prototype-based FIL method on CIFAR10, with results shown below ($\alpha=1.0$).
>
> |Metric|FCLPF|Li-FIL|
> |:-:|:-:|:-:|
> |$A(f)$|37.91|40.29|
> |$\bar{A}$|61.67|62.56|
>
> In addition, we have shown in Table 4 and Appendix C that without mixup, features become more vulnerable to inversion attacks. This suggests that limited feature diversity may introduce potential privacy risks, as prototype-based methods can approximate class-level averages for each client.
>
> Regarding scalability, we follow the dataset and experimental protocols in [2] to ensure fair comparison with prior work. Moreover, the complexity of the CVAE is primarily determined by the feature dimensionality rather than the number of classes. The class label is incorporated as a condition, and increasing the number of classes does not significantly affect the model size or training cost.
>
> ---
>
> > **Q2&W2.  Concerns about the Attack Mechanism and Privacy Guarantee**
>
> **R2:** We thank the reviewer for raising this important point. To provide a more comprehensive evaluation, we further conduct a Membership Inference Attack (MIA) experiment on CIFAR-10, which captures a complementary aspect of privacy risk (as space is limited, detailed configuration will be updated in the supplementary).
>
> |Metric|Raw|Mixup|DP|Li-FIL|
> |:-:|:-:|:-:|:-:|:-:|
> |Attack Acc.(%)|71.1|68.8|60.3|57.2|
>
> Together with feature inversion, MIA provides a more holistic assessment covering both visual leakage and membership inference risk. As shown, while DP already serves as a strong theoretical defense, Li-FIL consistently achieves lower attack success rates, indicating that Li-FIL offers additional protection beyond DP alone.
>
> Regarding the privacy guarantee, we clarify that our method provides *record-level local differential privacy (LDP)* for the uploaded features, as described in Sec. 3.2. Specifically, after $\ell_1$-norm clipping (which bounds the sensitivity by $\Delta\leq2C$), we apply the Laplace mechanism with noise scale $2C/\epsilon$. By standard DP results, each uploaded feature satisfies $\epsilon$-LDP, directly protecting the feature upload channel. A full-system $(\epsilon, \delta)$-DP characterization for the entire FL pipeline involves additional factors such as secure aggregation. These are complementary to our method and can be readily integrated with existing FL privacy frameworks.
>
> ---
> > **Q3&W3.  Systematic Analysis of the Communication Overhead**
>
> **R3:** We thank the reviewer for highlighting the importance of communication overhead. Li-FIL is inherently communication-efficient for two reasons:
> (1) we transmit **low-dimensional compact features** instead of raw images, reducing per-sample size from 3072 (CIFAR) to 512 dimensions ($\approx6\times$ smaller, up to $24\times$ for Tiny-Imagenet); (2) only high-confidence features are selected via a threshold $\tau$ and further reduced by mixup ($\approx50\%$). Therefore, the upload volume depends on the fraction of selected samples and **remains controllable and tunable** in practice.
>
> ---
> **References:**
>
> [1] Yoo M K, Park Y R. Federated class incremental learning: A pseudo feature based approach without exemplars[C]//Proceedings of the Asian Conference on Computer Vision. 2024: 488-498.
> [2] Li Y, Wang H, Xu W, et al. Unleashing the power of continual learning on non-centralized devices: A survey[J]. IEEE Communications Surveys & Tutorials, 2025.

---

> > ### Author Rebuttal · Reviewer_kfBT · 2026-04-04
> >
> > Thank you for your response. My concerns have been fully addressed.

---

### Official Review · Reviewer_urS8 · 2026-03-09

**Soundness:** 2
**Presentation:** 3
**Significance:** 2
**Originality:** 3
**Overall Recommendation:** 3
**Confidence:** 3

**Summary:**

This paper presents Li-FIL, a lightweight framework designed to address catastrophic forgetting, privacy leakage risks, and heavy communication/storage overheads in Federated Incremental Learning (FIL). The core contribution is the introduction of a decoupled replay mechanism that treats the encoder and classifier with distinct strategies. By deploying an extremely lightweight feature-level Conditional Variational Autoencoder (CVAE) on the server, Li-FIL replays feature streams of historical tasks instead of raw images. This approach effectively stabilizes classification decision boundaries while significantly reducing communication costs. Furthermore, the study establishes an enhanced privacy protection system by integrating Mixup feature perturbation and Differential Privacy (DP) on the client side, verified through feature inversion attack experiments. Overall, Li-FIL provides an efficient solution for continuous learning and knowledge retention in resource-constrained, privacy-sensitive edge environments, outperforming existing mainstream FIL methods across multiple benchmark datasets.

**Compliance With Llm Reviewing Policy:**

Affirmed.

**Key Questions For Authors:**

Q1. The paper integrates $\mathcal{L}_{task}$, $\mathcal{L}_{replay}$, $\mathcal{L}_{align}$, and $\mathcal{L}_{cont}$ in Section 3.4. How were the hyper-parameter weights between these loss terms set across all dataset experiments? Were they kept consistent? Could the authors provide a grid search or quantitative analysis of these coefficients to justify the rationale behind choosing specific weights?
Q2.Since the alignment and replay processes depend on a fixed feature dimension, how would the framework handle model heterogeneity? If some clients use ShuffleNet while others use ResNet, would it be necessary to introduce a Feature Projection Layer? How would this impact the alignment efficiency?
Q3.In the privacy-preserving experiments, how sensitive are the CVAE's training convergence and the final model performance to the privacy budget $\epsilon$? Specifically, under extreme noise levels (e.g., $\epsilon < 1$), can the local alignment loss $\mathcal{L}_{cont}$ based on contrastive learning still effectively extract discriminative features?

**Limitations:**

yes

**Strengths And Weaknesses:**

Strength
1. This paper breaks away from the traditional federated incremental learning framework that performs uniform updates on the entire model and innovatively proposes to decouple the learning strategies for the encoder and the classifier. By stabilizing decision boundaries through feature replay while constraining the representation space with alignment losses, it demonstrates strong academic insight.
2. The method in this paper does not require replaying original images; instead, it adopts a lightweight feature-level CVAE. This approach significantly reduces communication bandwidth (by approximately 6x) and drastically scales down the parameter size of the server-side generative model (by approximately 22x), making it highly suitable for resource-constrained edge computing scenarios.
3. The author provides a systematic privacy protection mechanism that integrates Mixup feature perturbation and Differential Privacy (DP) Laplace noise. The robustness against reconstruction attacks is quantitatively verified through feature inversion attack experiments, which is more rigorous than many existing FIL methods.
4. The paper conducts exhaustive comparisons on multiple mainstream datasets such as CIFAR-100 and Tiny-ImageNet against 8 SOTA baseline methods, providing strong empirical evidence of its superiority in average accuracy and forgetting mitigation.
Weakness
1. The methodology implicitly and strictly assumes that all participating clients must have exactly the same network architecture (such as ResNet-18 used in the experiment) to ensure feature space alignment. This greatly limits the universality of the method in actual heterogeneous FL environments where devices often possess diverse hardware capabilities and model structures.
2. The core of the Li-FIL framework relies on a composite loss function in $\mathcal{L}_{total} = \mathcal{L}_{task} + \lambda_{1}\mathcal{L}_{replay} + \lambda_{2}\mathcal{L}_{align} + \lambda_{3}\mathcal{L}_{cont}$. However, the authors did not provide any experimental basis or sensitivity analysis for the coefficients $\lambda_1, \lambda_2, \text{ and } \lambda_3$, making the robustness of the results across different task distributions questionable.
3. The current evaluation primarily focuses on 5-step or 10-step incremental processes. In long-term task streams, since the CVAE is trained on noisy features, the quality of generated representations may suffer from accumulated drift over time, yet this potential risk of "positive feedback error cascade" is not discussed.

---

> ### Author Rebuttal · Authors · 2026-03-30
>
> **Response to Reviewer urS8**
>
> We sincerely appreciate your review and insightful feedback.
>
> Due to a **markdown rendering issue on OpenReview**, we were unfortunately unable to view parts of the review. We have done our best to reconstruct the comments based on the available content. If we have misunderstood any points, we appreciate your clarification and will address them accordingly.
>
> Below are our point-by-point responses.
>
> ---
> > **Q1&W2. Concerns about the Hyper-parameters**
>
> **R1:** We thank the reviewer for this important point. We adopt a unified setting across all datasets with $λ_1=λ_2=λ_3=1/3$ and $β=1/2$, selected via validation-based grid search, as already briefly described in the **Configurations in Sec.4.1**. We provide a set of results on CIFAR10 below for multiple proportional configurations of the λ coefficients, where performance varies only within ±0.5–1%, indicating low sensitivity to these choices. From a design perspective, the equal weighting is motivated by the decoupled design of Li-FIL, where each term plays a complementary role. We will clarify this more explicitly in the final version.
>
> |Metric|2:1:1|1:2:1|1:1:2|1:1:1|
> |:-:|:-:|:-:|:-:|:-:|
> |$A(f)$|39.74|39.71|39.30|40.29|
> |$\bar{A}$|62.05|62.84|61.61|62.56|
> ---
> > **Q2&W1. Concerns about the Model Heterogeneity**
>
> **R2:** We thank the reviewer for this insightful question. In this work, we adopt a homogeneous backbone (ResNet-18) following [1] to ensure fair comparison with prior work, thus model heterogeneity is not explicitly addressed in the current version.
>
> For heterogeneous client encoders (e.g., ShuffleNet vs. ResNet), prior works [2] have demonstrated that introducing a lightweight Feature Projection Layer can map features into a shared latent space. To further validate the applicability of our method in heterogeneous settings, we conduct additional experiments on CIFAR-10 with a mixed setting of 50% ShuffleNet and 50% ResNet.
>
> |Metric|FedMD|ReFed|Li-FIL|
> |:-:|:-:|:-:|:-:|
> |$A(f)$|30.23|33.50|34.26|
> |$\bar{A}$|51.82|57.22|57.79|
>
> The effectiveness in heterogeneous settings can be attributed to (1) the CVAE modeling a global feature distribution across clients, (2) classifier replay, and (3) feature alignment objectives (MMD and contrastive alignment), which jointly regularize the projected features toward a consistent distribution.
>
> ---
> > **W3. Systematic Analysis of Positive Feedback Error Cascade**
>
> **R3:** We thank the reviewer for raising this insightful concern. We clarify that, consistent with [1], the 10-step incremental setting is already a challenging and commonly adopted benchmark.
>
> Importantly, Li-FIL does not rely on recursive self-training. The server-side CVAE is updated using **newly contributed features from current tasks**, rather than its own synthesized samples, thereby avoiding the self-reinforcing feedback loop described in the comment. Moreover, the uploaded features are carefully curated rather than noisy: (1) calibrated confidence-based filtering selects high-quality features, (2) mixup enriches and smooths the feature distribution, and (3) DP acts as a regularizer to mitigate client-specific bias. Together, these mechanisms maintain a stable feature pool and reduce the risk of long-term drift.
>
> To support this point, we conduct experiments on Tiny-ImageNet with long-term task streams (20 tasks).
>
> |Metric|FedAvg|GLFC|ReFed|Li-FIL|
> |:-:|:-:|:-:|:-:|:-:|
> |$A(f)$|14.29|17.84|16.90|18.77|
> |$\bar{A}$|20.31|22.89|21.71|24.02|
> ---
> > **Q3. Concerns about the Impact of the Privacy Budget ε**
>
> **R4:** Thank you for this insightful question. We agree that there is an inherent privacy–utility trade-off. We have already included the parameter sensitivity analysis of ε with the performance of Li-FIL under $ε=0.1$ in **Fig. 3**. The performance degrades as ε decreases, but the degradation is smooth, indicating that Li-FIL remains stable even under strong privacy constraints. Regarding convergence, stronger noise slows down training but does not prevent convergence. Empirically, the CVAE on average takes {15, 17, 24, 31} epochs to converge respectively for ε={10, 5, 1, 0.1}.
>
> To validate the effectiveness of $L_{cont}$ under extreme noise, we conduct an additional ablation on CIFAR10 with ε = 0.5. The results show that removing either $L_{cont}$ or $L_{align}$ degrades performance. This suggests that $L_{cont}$ can still extract discriminative features by leveraging top-$K$ neighbors instead of instance-level matching, improving robustness to noise.
>
> |Metric|Li-FIL|-w/o $L_{cont}$|-w/o $L_{align}$|
> |:-:|:-:|:-:|:-:|
> |$A(f)$|39.26|38.71|37.89|
> |$\bar{A}$|58.98|58.02|57.69|
> ---
> **References:**
>
> [1] Li Y, Wang H, Xu W, et al. Unleashing the power of continual learning on non-centralized devices: A survey[J]. IEEE Communications Surveys & Tutorials, 2025.
>
> [2] Li Y, Su H, Li H, et al. FedCD: Towards Consolidated Distillation for Heterogeneous Federated Learning[C]//Proceedings of the AAAI, 2026.

---

### Official Review · Reviewer_BazH · 2026-03-11

**Soundness:** 4
**Presentation:** 3
**Significance:** 4
**Originality:** 3
**Overall Recommendation:** 5
**Confidence:** 5

**Summary:**

To tackle the emerging challenges in Federated Incremental Learning (FIL) like resource efficiency, the authors proposed a Lightweight feature-level decoupled replay framework (Li-FIL). Leveraging the features synthesized by the generator on the server (a CVAE), Li-FIL performs decoupled replay to regularize the encoder and decoder individually to mitigate Catastrophic Forgetting. Local Differential Privacy (LDP) is further integrated into the framework to support the claim of privacy preservation. Extensive experiments show that Li-FIL outperforms SOTA methods in both effectiveness and efficiency.

**Compliance With Llm Reviewing Policy:**

Affirmed.

**Key Questions For Authors:**

Q1. Regarding W1, can provide more results on mentioned setting?
Q2. Regarding W2, can conduct experiment on synthesized features as in the curated features?

**Limitations:**

yes

**Strengths And Weaknesses:**

Strengths:
S1. The paper targets a real-world scenario in FIL where access to raw historical data is strictly prohibited on edge resource-constrained devices. The well-designed method mitigates communication, storage, and computational overhead during deployment.
S2. The writing and presentation of the paper is smooth.
S3. The core methodological contribution, a decoupled learning strategy for the encoder and classifier, is technically innovative, particularly in its use of both global and local feature alignment to regularize the encoder against Catastrophic Forgetting.
S4. Extensive results on four datasets compared with seven classic and strong baselines demonstrate Li-FIL's performance.

Major Weakness:
W1. My first concern is the incremental setting with at most 10 tasks and 20 classes per task on Tiny-Imagenet, which is quite limited. More challenging settings, for example, 20 tasks with 10 classes each, even 40 tasks should be considered.
W2. It seems that feature inversion is only performed on the uploaded features, is there a possibility that the features generated by CVAE can also be reconstructed, which will reveal sensitive information from the training data.

---

> ### Author Rebuttal · Authors · 2026-03-30
>
> **Response to Reviewer BazH**
>
> We sincerely appreciate your review and insightful feedback. Below, we provide detailed, point-by-point responses to each of your comments.
>
> ---
> > **Q1&W1.  Results on more Challenging Settings**
>
> **R1:** Thank you for raising this point regarding scalability. We clarify that, consistent with [1], the 10-step incremental setting is already a challenging and commonly adopted benchmark.
>
> To further evaluate the robustness of our method under more challenging and long-horizon incremental settings, we additionally conduct experiments on Tiny-ImageNet with increased numbers of tasks (20 tasks). These settings introduce significantly more severe catastrophic forgetting, and thus provide a stronger test of scalability in federated incremental learning.
>
> |Metric|FedAvg|GLFC|ReFed|Li-FIL|
> |:-:|:-:|:-:|:-:|:-:|
> |$A(f)$|14.29|17.84|16.90|18.77|
> |$\bar{A}$|20.31|22.89|21.71|24.02|
>
> From the results, we observe that Li-FIL consistently outperforms all baselines under these more challenging settings. This demonstrates that Li-FIL scales effectively to longer task sequences. We attribute this to the proposed **feature-based replay and decoupled optimization**, which together mitigate both decision-boundary drift and representation shift.
>
> ---
> > **Q2&W2.  Feature Inversion on Synthesized Features**
>
> **R2:** We thank the reviewer for this important concern. We note that synthesized features do **not** have a one-to-one correspondence with real data instances, as they are sampled from a learned feature distribution. Therefore, PSNR/SSIM-based inversion metrics, which require ground-truth images, are not directly applicable in this setting.
>
> Instead, we analyze this from a memorization perspective. The generator is trained only on privacy-preserving features (mixup + DP), and thus learns a smoothed distribution rather than memorizing instance-specific information. Moreover, our membership inference results (see **R3 to Reviewer MDXa**) show that Li-FIL significantly reduces distinguishability between training and non-training data, indicating weak memorization.
>
> Overall, since synthesized features are drawn from this privacy-preserving distribution, they do not correspond to any specific training instance and do not introduce additional privacy leakage.
>
> ---
> **Thank you again for the helpful feedback!**
>
> **References:**
>
> [1] Li Y, Wang H, Xu W, et al. Unleashing the power of continual learning on non-centralized devices: A survey[J]. IEEE Communications Surveys & Tutorials, 2025.

---

> > ### Author Rebuttal · Reviewer_BazH · 2026-04-02
> >
> > Thank you for your response. I have confirmed my score and have no further questions.

---

### Decision · Program_Chairs · 2026-04-30

**Decision:**

Accept (regular)

**Comment:**

The paper received overall positive reviews, with three accepts and one weak reject. This paper proposes Li-FIL, a lightweight federated incremental learning framework that performs feature-level decoupled replay using a server-side CVAE trained on privatized client features, with separate regularization strategies for the encoder and classifier to mitigate catastrophic forgetting while improving efficiency and privacy. Reviewers appreciated the practical importance of the problem, the clear and well-structured design, the originality of the decoupled replay mechanism, and the strong empirical performance across multiple benchmarks. The method was also viewed favorably for substantially reducing communication and generator complexity relative to image-level replay, while providing meaningful privacy protection through mixup and local differential privacy.
The main concerns centered on scalability to more challenging long-horizon settings and larger client populations, the choice of CVAE versus alternative generators, the scope of the privacy evaluation, and the dependence on homogeneous client architectures in the main experiments. Reviewers also asked for clearer hyperparameter sensitivity analysis, more systematic communication-cost discussion, and stronger justification that synthesized features do not introduce additional privacy leakage. These concerns were largely addressed in the rebuttal through added experiments on harder incremental settings and more clients, comparisons and discussion of alternative generator choices, membership inference evaluation, clarification of the local-DP guarantee on uploaded features, and additional evidence for robustness in heterogeneous model settings. While one reviewer remained somewhat unconvinced on some broader applicability issues, the overall consensus was that the paper makes a technically sound and practically valuable contribution to federated incremental learning. Therefore, the AC recommends accepting the paper.